# Differences between atrial fibrillation diagnosed before and after stroke: A large real-world cohort study

Yu-Kang Chang[1,2,3,4], Chih-Cheng Hsu[5,6], Chi-Ting Huang[5], Chi-Hsun Lien[2,7], Hung-Yi Hsu[2,7]*

1 Department of Medical Research, Tungs' Taichung MetroHarbor Hospital, Taichung, Taiwan,
2 Department of Life Science, College of Life Science, National Chung Hsing University, Taichung, Taiwan,
3 Department of Post-Baccalaureate Medicine, College of Medicine, National Chung Hsing University, Taichung, Taiwan, 4 Department of Nursing, Jenteh Junior College of Medicine, Nursing and Management, Miaoli, Taiwan, 5 Institute of Population Health Sciences, National Health Research Institutes, Miaoli, Taiwan, 6 National Center for Geriatrics and Welfare Research, National Health Research Institutes, Yunlin, Taiwan, 7 Department of Neurology, Tungs' Taichung MetroHarbor Hospital, Taichung, Taiwan

* hungyihsu@gmail.com

**Data Availability Statement:** Data are available from the National Health Insurance Research Database (NHIRD) published by Taiwan National

## Abstract

The clinical characteristics and long-term outcomes of patients with ischemic stroke (IS) and atrial fibrillation detected after stroke (AFDAS) have not been clearly established. Previous studies evaluating patients with AFDAS were limited by the low prescription rates of anticoagulants and short follow-up periods. Consecutive patients hospitalized for IS between 2014 and 2017 were identified from a National Health Insurance Research Database. The included patients were categorized into three groups: (1) known diagnosis of AF (KAF) before the index stroke, (2) AFDAS, and (3) without AF (non-AF). Univariable and multivariable Cox regression analyses were performed to estimate the hazard ratio (HR) for independent variables and recurrent IS, hemorrhagic stroke, or all-cause mortality. We identified 158,909 patients with IS of whom 16,699 (10.5%) had KAF and 7,826 (4.9%) had AFDAS. The patients with AFDAS were younger, more often male, and had lower $CHA_2DS_2$-VASc scores (3.8 ± 1.9 versus 4.9 ± 1.8, $p < 0.001$) than the patients with KAF. Anticoagulant treatment significantly reduced the risks of all outcomes. The standardized mortality rates were 40.4, 28.6, and 18.4 (per 100 person-years) for the patients with KAF, AFDAS, and non-AF, respectively. Compared with AFDAS, KAF was associated with lower risks of recurrent IS [hazard ratio (HR): 0.91, 95% confidence interval (CI): 0.86–0.97, $p < 0.01$] and hemorrhagic stroke (HR: 0.88, 95% CI: 0.79–0.99, $p < 0.01$) and a higher risk of all-cause mortality (HR: 1.11, 95% CI: 1.07–1.16, $p < 0.001$). The risks of recurrent IS and hemorrhagic stroke were higher and of all-cause mortality was lower for patients with AFDAS than with KAF. There is a strong need to refine treatment modalities to reduce the high mortality in patients with KAF and AFDAS.

Health Insurance (NHI) Bureau. Due to legal restrictions imposed by the government of Taiwan in relation to the "Personal Information Protection Act," data cannot be made publicly available. Requests for data can be sent as a formal proposal to the NHIRD (https://dep.mohw.gov.tw/DOS/cp-5119-59201-113.html). These are third party data, and we confirm that others would be able to access these data in the same manner as the authors and that the authors did not have any special access privileges that others would not have.

**Funding:** This work was funded by grants from Tungs' Taichung Metroharbor Hospital (protocol number TTMHH-109R0005). The funders had no role in study design, data collection and analysis, decision to publish, or preparation of the manuscript.

**Competing interests:** The authors declare no potential conflicts of interest with respect to the research, authorship, and/or publication of this article.

## Introduction

Atrial fibrillation (AF) related ischemic strokes (IS) are more often fatal or disabling than IS from other causes [1]. On the other hand, IS is a strong independent predictor of future stroke and death in patients with AF [2–4]. Patients with AF who ever experienced IS or transient ischemic attack (TIA) are at a higher risk of future adverse events. However, the long-term outcomes of the subgroup of AF patients with IS have not been well addressed in clinical studies.

Patients with IS may have had previously known AF (KAF) or have AF detected after experiencing a stroke (AFDAS). Substantial differences in comorbidities and outcomes might exist between KAF and AFDAS. Patients with AFDAS might have a lower risk of recurrent stroke and less cardiovascular comorbidity, including coronary heart disease, myocardial infarction, and heart failure than those of patients with KAF [5, 6]. Other studies have reported that AFDAS had outcomes and comorbidities similar to those of KAF [7] or had an even higher risk of death than that of KAF [8]. These discrepancies among different studies may be because of differences in populations, nomenclature of AF, methods of data collection, and diagnostic workups for AF detection. Moreover, previous studies were limited by a low rate of anticoagulation use and only a 1-year follow-up [5, 7, 9]. Only a few studies have reported the long-term outcomes of patients with IS and AF [8]. The differences in long-term outcomes between KAF and AFDAS need confirmation by more studies.

Current guidelines have suggested that anticoagulants, including vitamin-K antagonists and direct-acting oral anticoagulants (DOACs), effectively reduced the risks of IS and systemic embolization in patients with AF [4, 10, 11]. However, the prescription rates of anticoagulants in previous studies of patients with AFDAS were relatively low [5, 7, 9]. The benefits of anticoagulant treatment in AFDAS and the long-term prognosis need further investigation.

The aims of this study were to evaluate the clinical characteristics of AFDAS in patients with IS in a large population-based cohort; the differences in comorbidities and outcomes among the patients with KAF, AFDAS, and without AF; and the effects of anticoagulant treatment in patients with IS with either KAF or AFDAS.

## Methods

### Data resources

This study used data from the National Health Insurance (NHI) program, which was established in 1995 to offer comprehensive coverage for hospitalization and outpatient services and prescription medications to >99% of the residents of Taiwan. The data for this retrospective study were retrieved from the National Health Insurance Research Database (NHIRD) of the NHI program in Taiwan. We assessed the NHIRD in July 2021 for the first time. Since the data does not contain personally identifying information, the Institutional Review Board of this hospital determined after a thorough review that informed consent was unnecessary for this study.

### Study population

We identified adult patients who were ≧ 18 years old and was hospitalized for IS between 2014 and 2017. The diagnosis of IS was on the basis of the International Classification of Diseases (ICD), 9th and 10th editions, Clinical Modification (ICD-9-CM and ICD-10-CM) codes, including ICD-9-CM codes 433.x and 434.x as well as ICD-10-CM codes I63.x, I65.x, and I66.x as the principal discharge diagnosis. The AF diagnosis was based on the ICD-9-CM code 427.31 and ICD-10-CM codes I48.0, I48.1, I48.2, and I48.91. All included patients had a least 1-year follow-up data in the NHIRD. According to these patients' histories of AF diagnosis, we

categorized them into three groups: (1) KAF (patients with a known diagnosis of AF before the index stroke), (2) AFDAS (patients with no previous diagnosis of AF but AF detected during their hospital stay for the index stroke), and (3) non-AF (patients without AF). The first hospitalization for the IS during this period was designated as the index hospitalization. The admission date was defined as the index date.

### Independent variables

We collected demographic variables, including sex, age, and comorbidities, from the NHI program datasets. These comorbidities included hypertension, diabetes mellitus, hyperlipidemia, coronary artery disease, heart failure, peripheral vascular disease, chronic kidney disease, chronic obstructive pulmonary disease, and prior stroke/TIA. All of the main comorbidities were based on diagnosis codes (S1 Table) from at least one prior hospitalization or two prior outpatient visits during the 1-year period before the index hospitalization. The diagnostic accuracies of important comorbidities in NHIRD, such as hypertension, diabetes mellitus, heart failure, myocardial infarction, hyperlipidemia, and chronic obstructive pulmonary disease, have been previously validated [12, 13].

A modified Charlson Comorbidity Index score was used to summarize comorbidity [14]. Due to the unavailability of clinical stroke scales in the claims data, we employed the stroke severity index (SSI) that is a validated surrogate for assessing stroke severity [15]. The SSI proved particularly valuable for adjusting the case mix in mortality models involving patients with IS in instances in which a traditional clinical stroke scale was not accessible. The SSI comprises seven components that can be readily extracted from the index hospitalization claims. The SSI values have a strong correlation with National Institutes of Health Stroke Scale (NIHSS) scores [15] at admission and could be converted into estimated NIHSS scores [16]. Accordingly, the estimated NIHSS scores were categorized as mild ($<8$), moderate ($8-15$), or severe ($>15$) stroke, aligning with a previous study that used the NIHSS to prognosticate stroke outcomes. The $CHADS_2$ [17] and $CHA_2DS_2$-VASc scores [3] were also calculated for all patients. Antithrombotic treatments of study subjects were ascertained from NHI drug codes and Anatomical Therapeutic Chemical (ATC) codes. Antithrombotic prescriptions (including antiplatelet agents, warfarin and DOACs) were extracted during the follow-up period after the index stroke hospitalization until the study end date, outcome event, or death. Antithrombotic treatments of study subjects were divided into antiplatelet agents, warfarin, and DOACs groups on the basis of NHI drug codes and Anatomical Therapeutic Chemical codes.

### Outcome variables

We sought to evaluate outcomes encompassing recurrent IS, hemorrhagic stroke, or death. Mortality status was based on the cause of death data from NHI. Patient follow-up continued until any recurrence of IS, hemorrhagic stroke, death, or December 31, 2017, whichever came first.

### Statistical analysis

We presented continuous variables as means with standard deviations and presented categorical variables as counts with percentages. To compare baseline characteristics among the KAF, AFDAS, and non-AF groups, we used Chi-square tests for categorical variables and One-way analysis of variance for continuous variables. Univariable analysis was used to estimate the hazard ratio (HR) and 95% confidence interval (CI) for each independent variable. Multivariable Cox regression analyses were performed to estimate the HRs and 95% CIs for the association between AF subtype and the risk of recurrent ischemic stroke, hemorrhagic stroke, and all-cause mortality, adjusting for potential confounders including age, sex, stroke severity,

comorbidities, and antithrombotic treatment. We also conducted multivariable Cox regression analyses in the subpopulation of patients with AF (AFDAS and KAF groups combined) to evaluate the effectiveness of OAC therapy in reducing the risks of recurrent ischemic stroke, hemorrhagic stroke, and all-cause mortality. We ensured that all independent variables met the proportional hazards assumption by performing the Schoenfeld residuals test. All $p$-values were two-sided, and those <0.05 were accepted as indicating statistical significance. SAS 9.4 software (SAS Institute Inc., Cary, North Carolina, USA) was used to perform all statistical analyses.

## Results

Overall, 158,909 patients with IS were included in this study. Among these patients, 16,699 (10.5%) had KAF and 7,826 (4.9%) had AFDAS. The demographic characteristics of the different groups are presented in Table 1. Compared with the AFDAS patients, the KAF patients were older, more likely to be female, and had higher $CHA_2DS_2$-VASc scores and incidences of comorbidities, excluding hyperlipidemia. Compared with the non-AF group, the AFDAS group was older, had more female patients, and exhibited more comorbidities except for peripheral arterial disease and chronic obstructive pulmonary disease. There were no significant differences in the SSI scores and estimated NIHSS between the KAF and AFDAS groups. However, the non-AF group had lower $CHA_2DS_2$-VASc scores, SSI values, and estimated NIHSS scores than those in both the KAF and AFDAS groups.

The prescription rate of DOACs was higher in the AFDAS group (48.8%) than in the KAF (36.6%, $p < 0.001$). Significant differences were observed in all outcome measures (recurrent IS, hemorrhagic stroke, and all-cause mortality) among the three groups. The rates of recurrent IS and hemorrhagic stroke were highest in the AFDAS patients, whereas the all-cause mortality rate was highest in the KAF patients. The standardized mortality rates were 40.4, 28.6, and 18.4 (per 100 person-years) for the patients with KAF, AFDAS, and non-AF, respectively. In patients with KAF, 7304 (47.2%) patients had received OACs treatment within the 6 months preceding the index stroke. Adherence to OAC therapy was evaluated using the proportion of days covered (PDC), calculated by dividing the days a patient had medication by the days in the observation period. Patients with a PDC of ≥80% were considered adherent. Over the 6 months and 30 days prior to the index stroke, 4462 (26.7%) and 4068 (24.4%) patients, respectively, were adherent to OAC therapy.

The results of univariable and multivariable Cox regression models for predicting recurrent IS, hemorrhagic stroke, and all-cause mortality are presented in Tables 2–4, respectively. Risk factors for recurrent IS and all-cause mortality included male sex, age from 65–74 years, hypertension, diabetes mellitus, heart failure, prior stroke or TIA, severity of index stroke, KAF, and AFDAS (Tables 2 and 4). Male sex, age >75 years, severity of index stroke, KAF, and AFDAS increased the risk of recurrent hemorrhagic stroke. Coronary artery disease, prior stroke or TIA, and anticoagulant treatment reduced the risk of recurrent hemorrhagic stroke (Table 3). Anticoagulant treatment significantly reduced the risks of recurrent IS (HR: 0.35, 95% CI: 0.33–0.36, $p < 0.001$), hemorrhagic stroke (HR: 0.42, 95% CI: 0.38–0.45, $p < 0.001$), and all-cause mortality (HR: 0.51, 95% CI: 0.49–0.52, $p < 0.001$).

Table 5 presents the adjusted hazard ratios for the pairwise comparisons of recurrent ischemic stroke, hemorrhagic stroke, and all-cause mortality risks between the AF subtype groups. Compared to patients without AF, both AFDAS and KAF were associated with significantly higher risks of all outcomes. In the direct comparison between AFDAS and KAF, the patients with KAF had lower risks of recurrent IS (HR: 0.91, 95% CI: 0.86–0.97, $p < 0.01$) and hemorrhagic stroke (HR: 0.88, 95% CI: 0.79–0.99, $p < 0.01$) and a higher risk of all-cause mortality

**Table 1. Clinical characteristics, antithrombotic medications and outcomes of study populations.**

| | KAF, n(%) | AFDAS, n(%) | Non-AF, n(%) | p value |
|---|---|---|---|---|
| Sex (male) | 8414 (50.4) | 4148 (53.0)* | 81418 (60.6)*# | < .0001 |
| Age (year), mean (SD) | 77.8 (10.8) | 76.1 (11.6)* | 69.3 (13.9)*# | < .0001 |
| Age (year) | | | | < .0001 |
| <65 | 2207 (13.2) | 1421 (18.2) | 50185 (37.3) | |
| 65–74 | 3554 (21.3) | 1807 (23.1) | 32833 (24.4) | |
| 75+ | 10938 (65.5) | 4598 (58.8) | 51366 (38.2) | |
| Risk factors | | | | |
| Hypertension | 11336 (67.9) | 4607 (58.9)* | 74913 (55.8)*# | < .0001 |
| Diabetes mellitus | 5346 (32.0) | 1903 (24.3)* | 43706 (32.5)# | < .0001 |
| Hyperlipidemia | 3481 (20.9) | 1643 (21.0) | 31408 (23.4)*# | < .0001 |
| Coronary artery disease | 2750 (16.5) | 582 (7.4)* | 8826 (6.6)*! | < .0001 |
| Heart failure | 4907 (29.4) | 620 (7.9)* | 6529 (4.9)*# | < .0001 |
| Peripheral artery disease | 768 (4.6) | 210 (2.7)* | 3493 (2.6)* | < .0001 |
| Chronic kidney disease | 2088 (12.5) | 501 (6.4)* | 10129 (7.5)*# | < .0001 |
| Chronic obstructive pulmonary disease | 1719 (10.3) | 352 (4.5)* | 6387 (4.8)* | < .0001 |
| Prior stroke | 1261 (7.6) | 347 (4.4)* | 8894 (6.6)*# | < .0001 |
| Modified CCI score, mean (SD) | 1.2 (1.6) | 0.6 (1.2)* | 0.9 (1.5)*# | < .0001 |
| SSI, mean (SD) | 11.6 (6.4) | 11.4 (6.4) | 8.2 (5.5)*# | < .0001 |
| NIHSS | | | | < .0001 |
| <8 | 6113 (36.6) | 2862 (36.6) | 82128 (61.1) *# | |
| 8–15 | 3884 (23.3) | 1879 (24) | 26684 (19.9) *# | |
| >15 | 6702 (40.1) | 3085 (39.4) | 25572 (19.0) *# | |
| CHA$_2$DS$_2$-VASc, mean (SD) | 4.9 (1.8) | 3.8 (1.9)* | 3.4 (2.0)*# | < .0001 |
| CHA$_2$DS$_2$, mean (SD) | 3 (1.4) | 2.1 (1.3)* | 2.1 (1.5)*# | < .0001 |
| Antithrombotic treatment | | | | |
| Antiplatelet | 9295 (55.7) | 4417 (56.4) | 104451 (77.7)*# | < .0001 |
| Wafarin | 2505 (15.0) | 1085 (13.9)§ | 3576 (2.7)*# | < .0001 |
| DOAC | 6114 (36.6) | 3821 (48.8)* | 3183 (2.4)*# | < .0001 |
| Outcomes | | | | |
| Recurrent ischemic stroke | 3503 (21.0) | 1763 (22.5) @ | 24341 (18.1) *# | < .0001 |
| Intracranial hemorrhage | 890 (5.3) | 503 (6.4) * | 6517 (4.9) @# | < .0001 |
| All-cause mortality | 9282 (55.6) | 3492 (44.6) * | 41564 (30.9) *# | < .0001 |

KAF: known atrial fibrillation; AFDAS: atrial fibrillation detected after stroke; Non-AF: without atrial fibrillation; SD: standard deviation; CCI: Charlson comorbidity index; SSI: stroke severity index; NIHSS: national institutes of health stroke scale; DOACs: direct-acting oral anticoagulants.

*p<0.001 vs. KAF

@p <0.01 vs. KAF

# p<0.001 vs. AFDAS

§ p <0.05 vs. KAF

! P<0.01 vs NAF

(HR: 1.11, 95% CI: 1.07–1.16, $p < 0.001$). After accounting for the competing risk of death using the Cox regression model (S2 Table), the HR for recurrent ischemic stroke in KAF versus AFDAS was 0.87 (95% CI: 0.82–0.92, p<0.01), and the HR for hemorrhagic stroke was 0.91 (95% CI: 0.84–0.99, p<0.05), indicating that patients with KAF have a lower risk of both recurrent ischemic stroke and hemorrhagic stroke compared to those with AFDAS.

**Table 2. Univariable and multivariable Cox regression models to predict the recurrent ischemic stroke at the end of follow-up.**

|  | Univariable | | Multivariable* | |
|---|---|---|---|---|
|  | HR(95%CI) | *p* value | HR(95%CI) | *p* value |
| Sex (male vs. female) | 1.13(1.10–1.15) | < .0001 | 1.16(1.13–1.19) | < .0001 |
| Age |  |  |  |  |
| <65 | Ref. | Ref. | Ref. | Ref. |
| 65–74 | 1.09(1.06–1.13) | < .0001 | 1.07(1.04–1.10) | < .0001 |
| 75+ | 1.09(1.06–1.12) | < .0001 | 1.02(0.99–1.05) | 0.1506 |
| Risk factors |  |  |  |  |
| Hypertension (Yes vs. No) | 1.09(1.06–1.11) | < .0001 | 1.04(1.02–1.07) | 0.0010 |
| Diabetes mellitus (Yes vs. No) | 1.12(1.09–1.15) | < .0001 | 1.09(1.06–1.12) | < .0001 |
| Hyperlipidemia (Yes vs. No) | 1.05(1.02–1.08) | 0.0006 | 1.02(0.99–1.05) | 0.2792 |
| Coronary artery disease (Yes vs. No) | 1.16(1.11–1.21) | < .0001 | 1.07(1.02–1.11) | 0.0034 |
| Heart failure (Yes vs. No) | 1.15(1.10–1.20) | < .0001 | 1.06(1.01–1.11) | 0.0131 |
| Peripheral artery disease (Yes vs. No) | 1.19(1.11–1.27) | < .0001 | 1.12(1.05–1.19) | 0.0011 |
| Chronic kidney disease (Yes vs. No) | 1.02(0.97–1.07) | 0.3999 | 0.92(0.88–0.97) | 0.0007 |
| Prior stroke/TIA (Yes vs. No) | 1.09(1.05–1.14) | < .0001 | 1.06(1.01–1.11) | 0.0091 |
| Modified CCI score | 1.03(1.02–1.04) | < .0001 | 1.00(0.99–1.01) | 0.5628 |
| SSI | 1.071(1.068–1.073) | < .0001 | 1.070(1.068–1.072) | < .0001 |
| Anticoagulant treatment (Yes vs. No) | 0.56(0.54–0.59) | < .0001 | 0.35(0.33–0.36) | < .0001 |
| AF subtype |  |  |  |  |
| Non-AF | Ref. | Ref. | Ref. | Ref. |
| AFDAS | 1.33(1.29–1.38) | < .0001 | 2.37(2.25–2.50) | < .0001 |
| KAF | 1.34(1.27–1.41) | < .0001 | 2.03(1.95–2.11) | < .0001 |

* Multivariable models adjusted for potential confounders including age, sex, stroke severity index score, comorbidities (hypertension, diabetes, hyperlipidemia, coronary artery disease, heart failure, peripheral artery disease, chronic kidney disease, prior stroke/TIA), modified Charlson Comorbidity Index score, and anticoagulant treatment.

KAF: known atrial fibrillation; AFDAS: atrial fibrillation detected after stroke; Non-AF: without atrial fibrillation; TIA: transient ischemic attack; CCI: Charlson comorbidity index; SSI: stroke severity index. Ref.: reference

Cox regression analysis of the effectiveness of OAC therapy in only patients with AF showed that OAC therapy was associated with significantly lower risks of recurrent ischemic stroke (HR: 0.20, 95% CI: 0.19–0.21, p<0.001), hemorrhagic stroke (HR: 0.20, 95% CI: 0.18–0.22, p<0.001), and all-cause mortality (HR: 0.26, 95% CI: 0.25–0.27, p<0.001) compared to no OAC therapy, after adjusting for potential confounders (S3 Table). The benefits of OAC therapy were similar in both KAF and AFDAS subgroups (S4 and S5 Tables). DOACs were associated with significantly lower risks of hemorrhagic stroke compared to warfarin in KAF subgroup (HR 0.49, 95% CI: 0.35–0.69, p<0.001) and AFADS subgroup (HR 0.47, 95% CI: 0.27–0.81, p<0.001). The risks of hemorrhagic stroke were not significantly different between Warfarin and antiplatelet treatment in both KAF and AFDSA subgroups.

## Discussion

This study evaluated the clinical characteristics and outcomes of patients with IS and KAF, AFDAS, and non-AF in a real-world Chinese cohort. The large patient population and longer follow-up period provided valuable information that will contribute to a better understanding of the possible effects of AF in patients with IS and of the potential interventions for these patients. In our cohort, the incidence of severe stroke was twice as frequent in the patients with AF than in the patients without AF. Previous studies also showed that AF-associated

**Table 3. Univariable and multivariable Cox regression models to predict the hemorrhage stroke at the end of follow-up.**

| | Univariable | | Multivariable* | |
|---|---|---|---|---|
| | HR(95%CI) | *p* value | HR(95%CI) | *p* value |
| Sex (male vs. female) | 1.19(1.13–1.25) | < .0001 | 1.17(1.12–1.23) | < .0001 |
| Age | | | | |
| <65 | Ref. | Ref. | Ref. | Ref. |
| 65–74 | 0.94(0.87–0.99) | 0.0365 | 0.95(0.90–1.01) | 0.0756 |
| 75+ | 0.89(0.84–0.93) | < .0001 | 0.87(0.83–0.92) | < .0001 |
| Risk factors | | | | |
| Hypertension (Yes vs. No) | 0.92(0.88–0.97) | 0.0007 | 0.98(0.94–1.03) | 0.4738 |
| Diabetes mellitus (Yes vs. No) | 0.95(0.90–0.99) | 0.0378 | 1.00(0.95–1.06) | 0.9709 |
| Hyperlipidemia (Yes vs. No) | 0.92(0.87–0.97) | 0.0015 | 0.97(0.91–1.02) | 0.2334 |
| Coronary artery disease (Yes vs. No) | 0.89(0.82–0.98) | 0.0185 | 0.89(0.81–0.97) | 0.0121 |
| Heart failure (Yes vs. No) | 0.99(0.90–1.09) | 0.8578 | 0.99(0.90–1.09) | 0.8492 |
| Peripheral artery disease (Yes vs. No) | 0.91(0.78–1.05) | 0.1961 | 0.95(0.82–1.10) | 0.4753 |
| Chronic kidney disease (Yes vs. No) | 0.99(0.91–1.09) | 0.8776 | 1.03(0.94–1.13) | 0.5483 |
| Prior stroke/TIA (Yes vs. No) | 0.92(0.85–1.00) | 0.0623 | 0.79(0.72–0.87) | < .0001 |
| Modified CCI score | 0.98(0.97–0.99) | 0.0017 | 0.99(0.97–1.01) | 0.3651 |
| SSI | 1.069(1.065–1.073) | < .0001 | 1.073(1.069–1.077) | < .0001 |
| Anticoagulant treatment (Yes vs. No) | 0.66(0.61–0.71) | < .0001 | 0.42(0.38–0.45) | < .0001 |
| AF subtype | | | | |
| Non-AF | Ref. | Ref. | Ref. | Ref. |
| AFDAS | 1.26(1.17–1.35) | < .0001 | 2.43(2.20–2.69) | < .0001 |
| KAF | 1.43(1.30–1.57) | < .0001 | 2.06(1.90–2.24) | < .0001 |

* Multivariable models adjusted for potential confounders including age, sex, stroke severity index score, comorbidities (hypertension, diabetes, hyperlipidemia, coronary artery disease, heart failure, peripheral artery disease, chronic kidney disease, prior stroke/TIA), modified Charlson Comorbidity Index score, and anticoagulant treatment.

KAF: known atrial fibrillation; AFDAS: atrial fibrillation detected after stroke; Non-AF: without atrial fibrillation; TIA: transient ischemic attack; CCI: Charlson comorbidity index; SSI: stroke severity index. Ref.: reference

strokes tended to be more severe and debilitating [4, 18]. The stroke severity indices were not different between KAF and AFDAS. Compared with non-AF, both KAF and AFDAS were associated with increased risks of recurrent IS, future hemorrhagic stroke, and all-cause mortality even after adjusting for traditional risk factors.

The incidence of all-cause mortality was significantly higher in the KAF group than in the AFDAS group. It is sensible because the patients with KAF had higher $CHA_2DS_2$-VASc scores and more cardiovascular comorbidities than those of the patients with AFDAS. An early study in the same region of Taiwan also found that the 1-year mortality rate was significantly higher in patients with KAF than in patients with AFDAS [19]. A hospital-based cohort showed that the in-hospital mortality rate was approximately 15% in the KAF and AFDAS groups of patients [20]. An inception cohort showed a 12.2% per year mortality rate in KAF and a 15.8% per year mortality rate in AF diagnosed within 6 months after index stroke [8]. The discrepancies in the studied populations and clinical evaluations made the comparisons among studies difficult.

In the present study, the risks of recurrent IS and hemorrhagic stroke were higher in the patients with AFDAS than in the patients with KAF and patients without AF. The differences in risk factors and comorbidities between KAF and AFDAS were unlikely to contribute to the increases in recurrent IS and hemorrhagic stroke in AFDAS. Compared with the AFDAS

**Table 4. Univariable and multivariable Cox regression models to predict the all-cause mortality at the end of follow-up.**

| | Univariable | | Multivariable* | |
|---|---|---|---|---|
| | HR(95%CI) | *p* value | HR(95%CI) | *p* value |
| Sex (male vs. female) | 0.83(0.81–0.85) | < .0001 | 1.06(1.05–1.08) | < .0001 |
| Age | | | | |
| <65 | Ref. | Ref. | Ref. | Ref. |
| 65–74 | 1.60(1.54–1.66) | < .0001 | 1.59(1.54–1.63) | < .0001 |
| 75+ | 3.81(3.70–3.92) | < .0001 | 3.45(3.36–3.53) | < .0001 |
| Risk factors | | | | |
| Hypertension (Yes vs. No) | 1.32(1.29–1.35) | < .0001 | 0.93(0.91–0.95) | < .0001 |
| Diabetes mellitus (Yes vs. No) | 1.29(1.27–1.32) | < .0001 | 1.02(1.00–1.05) | 0.0236 |
| Hyperlipidemia (Yes vs. No) | 0.79(0.77–0.81) | < .0001 | 0.74(0.74–0.78) | < .0001 |
| Coronary artery disease (Yes vs. No) | 1.58(1.53–1.64) | < .0001 | 1.01(0.98–1.04) | 0.4948 |
| Heart failure (Yes vs. No) | 2.63(2.55–2.71) | < .0001 | 1.44(1.41–1.48) | < .0001 |
| Peripheral artery disease (Yes vs. No) | 1.69(1.60–1.78) | < .0001 | 1.09(1.04–1.13) | 0.0002 |
| Chronic kidney disease (Yes vs. No) | 2.34(2.27–2.41) | < .0001 | 1.36(1.33–1.40) | < .0001 |
| Prior stroke/TIA (Yes vs. No) | 1.33(1.28–1.37) | < .0001 | 0.90(0.87–0.92) | < .0001 |
| Modified CCI score) | 1.208(1.202–1.213) | < .0001 | 1.19(1.18–1.19) | < .0001 |
| SSI | 1.182(1.180–1.184) | < .0001 | 1.169(1.167–1.171) | < .0001 |
| Anticoagulant treatment (Yes vs. No) | 0.74(0.72–0.77) | < .0001 | 0.51(0.49–0.52) | < .0001 |
| AF subtype | | | | |
| Non-AF | Ref. | Ref. | Ref. | Ref. |
| AFDAS | 2.29(2.23–2.35) | < .0001 | 1.97(1.90–2.05) | < .0001 |
| KAF | 1.61(1.54–1.68) | < .0001 | 1.93(1.88–1.99) | < .0001 |

* Multivariable models adjusted for potential confounders including age, sex, stroke severity index score, comorbidities (hypertension, diabetes, hyperlipidemia, coronary artery disease, heart failure, peripheral artery disease, chronic kidney disease, prior stroke/TIA), modified Charlson Comorbidity Index score, and anticoagulant treatment.

KAF: known atrial fibrillation; AFDAS: atrial fibrillation detected after stroke; Non-AF: without atrial fibrillation; TIA: transient ischemic attack; CCI: Charlson comorbidity index; SSI: stroke severity index. Ref.: reference

group, the KAF group had a 1.5% absolute lower incidence of recurrent IS and an 11.0% absolute higher incidence of all-cause mortality. The high mortality rate in the patients with KAF could be a competing risk for recurrent IS and hemorrhagic stroke. However, the results of the competing risk analysis (S2 Table) showed that the higher mortality rate in the KAF group did not substantially bias the comparison of stroke risks between the AFDAS and KAF groups. The reasons behind the elevated risk of recurrent strokes in patients with AFDAS warrant further investigation to better understand the pathophysiology and guide targeted interventions.

In addition, the underlying mechanisms of recurrent strokes in patients with AFDAS may be heterogeneous. Patients with neurogenic AF, who developed AF resulting from a stroke, may have non-cardioemoblic strokes. Similarly, some recurrent cerebrovascular events in patients with AFDAS may be related to non-cardioembolic pathways, such as atherosclerosis or small vessel disease, rather than AF itself. Distinguishing the proportion of AFDAS patients with a true cardioembolic genesis of their neurological events remains a major challenge. Nonetheless, our findings suggest that AFDAS patients represent a high-risk group that warrants close monitoring and aggressive secondary prevention measures, regardless of the precise etiology of their AF or stroke. Future studies incorporating these additional diagnostic modalities are needed to optimize risk stratification tools and therapeutic approaches.

**Table 5. Multivariable Cox regression models to predict the outcome of recurrent ischemic stroke, hemorrhage stroke, or death at the end of follow-up.**

| Outcome Measures | Non-AF | AFDAS | KAF |
|---|---|---|---|
| Recurrent ischemic stroke | | | |
| n(%) | 24341(18.1) | 1763(22.5) | 3503(21.0) |
| HR (95%CI)# | Ref. | 1.81(1.71–1.90)** | 1.65(1.59–1.72)** |
| HR (95%CI)# | 0.55(0.53–0.58)** | Ref. | 0.91(0.86–0.97)* |
| Hemorrhage stroke | | | |
| n(%) | 6517(4.8) | 503(6.4) | 890(5.3) |
| HR (95%CI)# | Ref. | 1.88(1.70–2.08)** | 1.66(1.53–1.80)** |
| HR (95%CI)# | 0.53(0.48–0.59)** | Ref. | 0.88(0.79–0.99)* |
| Death | | | |
| n(%) | 41564(30.9) | 3492(44.6) | 9282(55.6) |
| HR (95%CI)# | Ref. | 1.27(1.22–1.31)** | 1.41(1.37–1.45)** |
| HR (95%CI)# | 0.79(0.76–0.82)** | Ref. | 1.11(1.07–1.16)** |

KAF: known atrial fibrillation; AFDAS: atrial fibrillation detected after stroke; Non-AF: without atrial fibrillation; CI: confidence interval. Ref.: reference.

#Adjusted age, sex, stroke severity index score, comorbidities (hypertension, diabetes, hyperlipidemia, coronary artery disease, heart failure, peripheral artery disease, chronic kidney disease, prior stroke/TIA), modified Charlson Comorbidity Index score, and anticoagulant treatment.

*P<0.01

**P<0.001.

The high mortality rates in the patients with KAF and patients with AFDAS in our real-world cohort are noteworthy. Previous cohorts of Chinese patients with IS and AF reportedly had 1-year mortality rates ranging from 15.7% to 22.1% [7, 19]. Longer follow-up periods and older age might have contributed to the high mortality rates in our cohort. The relatively high incidence of heart failure in our patients with KAF may also have contributed to the high mortality rate. In a meta-analysis of 71,683 patients from four randomized trials of DOACs, the adjusted mortality rate was 4.72% per year [21]. Pooled data from prospective cohorts showed a 10.2% per year mortality rate during 6,128 patient-years of follow-up in 5,314 patients [22]. The mortality rates were higher in real-world cohorts than the rates in randomized trials. Compared with patients with AFDAS, patients without stroke but with newly diagnosed non-valvular AF had an all-cause mortality rate of 3.83 per 100 person-years in the global observational GARFIELD-AF study [23]. The patients with AF and IS seemed more vulnerable than AF patients without stroke. There is a strong need to refine treatment modalities to improve post-stroke care and reduce stroke-related mortality. Previous studies suggested that the majority of deaths in patients with AF were attributable to cardiovascular etiologies [18, 21, 24]. Rhythm control with either AF ablation or antiarrhythmic drug therapy might decrease cardiovascular deaths in patients with either KAF or AFDAS [25, 26]. Further studies to investigate the long-term outcomes and causes of death in patients with stroke and AF are necessary to design effective interventions to reduce the high mortality in the population.

The prevalences of AFDAS (4.9%) and KAF (10.5%) in our cohort were similar to those in previous stroke registries [7, 27]. An early study that used the similar patient screening criteria and Taiwan NHIRD found only 481 patients with KAF and 680 patients with NAF between 2000 and 2012 [19]. Improved awareness of AF could contribute to the marked increase in AF diagnosis in patients with IS in the mid-2010s in Taiwan. Previous studies have found that the incidence of post-stroke AF varied from 3.2% to 24% [28]. Patients with acute IS reportedly

have in-hospital AFDAS rates between 11.5% and 12.8% in the western world [20, 29, 30]. Because IS is more frequently caused by small-vessel diseases and intracranial arterial stenosis in the Chinese population [27, 31], ethnic differences might have contributed to the low incidences of AFDAS and KAF in our cohort. A meta-analysis showed that AFDAS was detected in 5.1% of patients with IS during monitoring of <72 hours, whereas the rate of AFDAS increased to 15% with monitoring of >7 days [32]. The percentage of AFDAS in our cohort was relatively low, probably because of ethnic differences and infrequent use of long-term cardiac monitoring in real-world practice. However, the vast majority of AFDAS diagnoses could be made by 12-lead EKG or standard rhythm monitoring (telemetry and/or rhythm strips) [20]. Prolongation of cardiac monitoring might serve to detect less clinically relevant AF in low-risk populations [33]. Current guidelines suggest the use of long-term cardiac monitoring in selected patients only [4, 34].

In the present study, the patients with AFDAS included those who had preexisting AF but undiagnosed before the index stroke and those who had newly developed AF after a stroke, so-called neurogenic AF. Strictly speaking, neurogenic AF is AF diagnosed through long-term cardiac monitoring in patients with a cryptogenic stoke after initial evaluation [35]. Neurogenic AF, which is caused by hyperactivity and an imbalanced central nervous system, systemic inflammation, and endothelial dysfunction caused by stroke [35, 36], has been hypothesized to be a relatively benign low-burden AF. The rate of AFDAS was higher in our cohort than in previous study that used a restricted definition for diagnosis [5]. Compared with the non-AF group, the AFDAS group in our cohort had higher stroke severity, older age, and higher incidences of coronary artery disease and heart failure. The differences in clinical characteristics and outcomes between the AFDAS and non-AF groups suggested that the majority of our patients with AFDAS might have cardiac myopathy and preexisting AF before the index stroke rather than neurogenic AF. The high incidence of recurrent IS and hemorrhagic stroke in our AFDAS patients indicated that AFDAS was not a benign condition. Raising awareness of AF in communities and implementing standardized evaluations in high-risk patients to detect AF before stroke occurrence are essential for adequate interventions to decrease the AF-related morbidity and future devastating events [26, 37].

Anticoagulants were used in 51.6% of the KAF patients and 62.7% of the AFDAS patients in our cohort, and the prescription rates of anticoagulants were higher than those in previous studies in Chinese patients with IS and AF [7, 19]. The rates of recurrent strokes were high in our study because of longer follow-up periods and higher $CHA_2DS_2$-VASc scores. Our study demonstrates the benefits of OAC therapy in reducing the risks of recurrent ischemic stroke, hemorrhagic stroke, and all-cause mortality in patients with AF, including both AFDAS and KAF groups. These findings highlight the importance of prompt initiation and maintenance of OAC treatment in this high-risk population, in line with current guidelines. Although the AF patients with higher risk of bleeding complications probably were not taking anticoagulants in our cohort, which would create selection bias, anticoagulants use in our patients with AF was beneficial without an increased risk of hemorrhagic stroke in this large real-world cohort. In addition, our results showed that DOACs have a superior safety profile with regard to hemorrhagic stroke risk compared to warfarin and antiplatelet therapy in patients with AF, whereas warfarin treatment was not associated with a higher risk of hemorrhagic stroke compared to antiplatelet treatment. High bleeding risks should not be a reason to withhold anticoagulants [11].

Only about a quarter of our patients with KAF were adherent to OAC therapy before the occurrence of the index stroke. A post-hoc analysis from GLORIA-AF Registry (November 2011-December 2016) showed that the prescription rate (57.3%) in Asian patients with newly diagnosed AF was similar to our cohort [38]. Asian patients excluding Japanese patients were less likely prescribed an OAC compared to non-Asian individuals [38]. Although PDC

provides an indirect measure of treatment adherence, our findings suggest that a substantial proportion of KAF patients were not receiving guideline-recommended OAC therapy before their stroke events. The low rates of OAC prescription and adherence in this population underscore the urgent need for improved OAC initiation and adherence in clinical practice to prevent disabling strokes in high-risk patients with KAF.

Female patients with AF had a higher risk of IS and systemic embolism [39–41]. Although patients with KAF and AFDAS were more likely to be female in this cohort, multivariate regression analysis showed that male gender was associated with higher risks of adverse outcomes. Previous studies showed that female sex was a risk factor for stroke and thromboembolism in Western populations [41, 42], but was less evident in Asian populations [43, 44]. However, many confounders could not be evaluated in our retrospective cohort. The sex-related risks of adverse outcomes in patients with IS and AF need further prospective studies.

This study had several limitations. The topography of brain infarcts and the mechanisms of strokes could not be specified because we were unable to review electrocardiography and neuroimaging findings. Misclassification bias may occur; for example, AF patients might have lacunar infarction or ischemic stroke due to severe stenosis in a corresponding large artery or hypotension. The adherence to anticoagulants was assessed indirectly through dispensed prescriptions. The reasons for withholding anticoagulants in patients with AF were unknown. Those who did not take anticoagulants might have more comorbidity or be more vulnerable to bleeding complications. Hemorrhagic complications other than intracerebral hemorrhage were not evaluated. The workups for detecting AF were not standardized. The AFDAS group only included patients diagnosed with AF during the index hospitalization for stroke in this study. Patients who had paroxysmal AF or delayed onset AF after discharge may be classified as non-AF. Recent studies using continuous cardiac monitoring devices have shown that a significant proportion of patients with cryptogenic stroke may have undetected AF episodes in the weeks and months following the initial event [29, 45, 46]. The limited use of long-term cardiac monitoring probably underestimated the true prevalence and impact of AF in this population. Future studies using prolonged cardiac monitoring techniques are needed to more accurately characterize the incidence and timing of AF in patients with stroke, and to evaluate the implications for risk stratification and management. AF could be under-reported because the diagnosis of AF was based on claims data. Consequently, the possibility of incomplete clinical coding in some patients could not be excluded. The $CHA_2DS_2$-VASc and SSI scores, which were calculated retrospectively, could have been underestimated. Validated equations were used to derive the severity of stroke from claims data rather than directly from the primary record. Therefore, the generalizability of our results to different ethnic populations is unknown and requires additional studies.

In conclusion, the risks of recurrent IS and hemorrhagic stroke were higher but the risk of all-cause mortality was lower in the patients with AFDAS than in the patients with KAF. Anticoagulant treatment was found to reduce the risks of recurrent IS, hemorrhagic stroke, and all-cause mortality. Effective interventions to reduce the high all-cause mortality in patients with KAF and AFDAS require further investigation.

## Supporting information

**S1 Table. ICD-9-CM and ICD-10-CM codes for diagnosis of risk factors and comorbidities.** (DOCX)

**S2 Table. Multivariable Cox regression models to predict the outcome of recurrent ischemic stroke and hemorrhage stroke at the end of follow-up.** (DOCX)

**S3 Table. Multivariable Cox regression models to predict the outcome of recurrent ischemic stroke, hemorrhage stroke, or death at the end of follow-up in AF patients with and without OAC therapy.**
(DOCX)

**S4 Table. Multivariable Cox regression models to predict the outcome of recurrent ischemic stroke, hemorrhage stroke, or death at the end of follow-up in KAF patients with and without OAC therapy.**
(DOCX)

**S5 Table. Multivariable Cox regression models to predict the outcome of recurrent ischemic stroke, hemorrhage stroke, or death at the end of follow-up in AFDAS patients with and without OAC therapy.**
(DOCX)

## Acknowledgments

First, we would like to express our gratitude to everyone who supported us throughout the course of this research. We are also to thank the research team of National Health Research Institutes for their contribution. Finally, we would like to thank the participants for their contribution in the study from Tungs' Taichung Metroharbor Hospital.

## Author Contributions

**Data curation:** Chi-Ting Huang.

**Formal analysis:** Chi-Ting Huang.

**Investigation:** Hung-Yi Hsu.

**Methodology:** Yu-Kang Chang, Hung-Yi Hsu.

**Validation:** Yu-Kang Chang, Chih-Cheng Hsu, Chi-Hsun Lien.

**Writing – original draft:** Hung-Yi Hsu.

**Writing – review & editing:** Yu-Kang Chang, Chih-Cheng Hsu, Chi-Hsun Lien.

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
