## [Decision Letter · Decision Letter 0]

10 Apr 2024

PONE-D-23-43796Differences between atrial fibrillation diagnosed before and after stroke: a large real-world cohort studyPLOS ONE

Dear Dr. Hsu,

Thank you for submitting your manuscript to PLOS ONE. After careful consideration, we feel that it has merit but does not fully meet PLOS ONE’s publication criteria as it currently stands. Therefore, we invite you to submit a revised version of the manuscript that addresses the points raised during the review process.

We look forward to receiving your revised manuscript.

Kind regards,

Giulio Francesco Romiti

Academic Editor

PLOS ONE

Journal Requirements:

This work was funded by grants from Tungs’ Taichung Metroharbor Hospital (protocol number TTMHH-109R0005).

This work was funded by grants from Tungs’ Taichung Metroharbor Hospital (protocol number TTMHH-109R0005).

This work was funded by grants from Tungs’ Taichung Metroharbor Hospital (protocol number TTMHH-109R0005).

6. We notice that your supplementary tables are included in the manuscript file. Please remove them and upload them with the file type 'Supporting Information'. Please ensure that each Supporting Information file has a legend listed in the manuscript after the references list.

Reviewers' comments:

Reviewer's Responses to Questions

**Comments to the Author**

1. Is the manuscript technically sound, and do the data support the conclusions?

Reviewer #1: Partly

Reviewer #2: Partly

2. Has the statistical analysis been performed appropriately and rigorously? 

Reviewer #1: No

Reviewer #2: Yes

3. Have the authors made all data underlying the findings in their manuscript fully available?

Reviewer #1: No

Reviewer #2: Yes

4. Is the manuscript presented in an intelligible fashion and written in standard English?

Reviewer #1: Yes

Reviewer #2: Yes

5. Review Comments to the Author

**Reviewer #1:** In this study, the authors retrospectively investigated more than 160.000 thousand patients from from the National Health Insurance Research Database (NHIRD) of the NHI program in Taiwan. The aims of this study were to evaluate the clinical characteristics of newly diagnosed atrial fibrillation after a stroke (AFADS) in patients with ischemic stroke (IS) in a large population-based cohort; the differences in comorbidities and outcomes among the patients with known AF (KAF), AFADS, and without AF; the effects of anticoagulant treatment in patients with IS with either KAF or AFADS. The authors reported a very high mortality rate among patients with AF (both groups), and a higher risk of recurrent IS and hemorrhagic stroke for patients with KAF compared with AFADS.

The study is conducted in a large population and is overall well written, applying also a quite good statistical method. However, I have some major and minor comments:

Major

1) It is not clear at which time the data regarding OAC therapy has been extracted from the registry. Is it after the stroke events? Please provide this data.

2) Following the previous point, many patients with KAF should have been already treated with OAC therapy before the stroke. I believe that it would be very interesting to know how many of them were already on treatment before the stroke.

3) Since data were collected regarding antiplatelet therapy, direct oral anticoagulants (DOAC) and vitamin K antagonist (VKAs), it would be interesting to see the results of the cox regression analysis also for the subgroups of antithrombotic treatment, especially in regard of hemorragic events.

4) Regarding the Cox regression analysis, the authors did not report about the adjustments made in the multivariable analysis for the risk of recurrent stroke, hemorragic stroke and all-cause mortality (Tables 2,3 and 4). Please provide this data.

5) Since the independent variable including the type of AF (eg. Non-AF, AFADS, KAF) was already studied in the different models for the various endpoints (results reported in table 2, 3 and 4), I do not understand why it has been done the same thing also in table 5. Furthermore, the results of the analysis using Non-AF as reference reported in table 5 are different compared to those in Tables 2,3 and 4, suggesting different adjustments. Please specify.

6) Since the mortality rate for KAF is particularly high, the risk of competing events strongly limits the results regarding the recurrence of Ischemic stroke and hemorragic stroke. I suggest that reporting “Compared with AFADS, the patients with KAF had lower risks of recurrent IS (HR: 0.91, 95% CI: 0.86–0.97, p < 0.01) and hemorrhagic stroke (HR: 0.88, 95% CI: 0.79–0.99, p < 0.01)” may be misleading. I also suggest to provide potential strategies for mitigating the impact of competing events on the study outcomes, thus strengthening the validity and reliability of the results of the study.

7) The authors also reported that OAC therapy reduced the risk of all the outcomes of the study. However, they do not provide data regarding the population in which OAC therapy was studied. The results reported in the tables, suggest that they included in the cox regression analysis the whole population, also those with Non-AF, for which OAC therapy may not be indicated or beneficial. Please provide this data.

Minor

1) Discussion paragraph, page 8. “The incidences of AFADS (4.9%) and KAF (10.5%)…”. Maybe, it was meant “The prevalence”?

2) I also suggest in the discussion in the paragraph regardin OAC therapy to refer to this relevant paper on the argument “ Romiti GF, Corica B, Proietti M, Mei DA, Frydenlund J, Bisson A, Boriani G, Olshansky B, Chan YH, Huisman MV, Chao TF, Lip GYH; GLORIA-AF Investigators. Patterns of oral anticoagulant use and outcomes in Asian patients with atrial fibrillation: a post-hoc analysis from the GLORIA-AF Registry. EClinicalMedicine. 2023 Aug 25;63:102039. doi: 10.1016/j.eclinm.2023.102039. PMID: 37753446”.

3) Also, it may be interesting to refer to some gender disparities regarding the risk of adverse events. Here I provide a relevant article on the topic “Bucci T, Shantsila A, Romiti GF, Teo WS, Park HW, Shimizu W, Mei DA, Tse HF, Proietti M, Chao TF, Lip GYH; Asia-Pacific Heart Rhythm Society Atrial Fibrillation Registry Investigators. Sex-related differences in presentation, treatment, and outcomes of Asian patients with atrial fibrillation: a report from the prospective APHRS-AF Registry. Sci Rep. 2023 Oct 26;13(1):18375. doi: 10.1038/s41598-023-45345-3. PMID: 37884587; PMCID: PMC10603128.”

**Reviewer #2**: Dear Authors, thank you for submitting your valuable study.

You have the unquestionable merit of having addressed a still unclear subject, the management of which is still too arbitrary and burdened by mortality and morbidity rates still too high.

However, I have some suggestions to offer you to refine the article and enrich the discussion section:

- consideration should be given to the possibility that the results of the higher incidence of recurrent ischemic or hemorrhagic stroke in patients with AFDAS may be linked to the fact that atrial fibrillation in this cohort may recognize different etiopathogenic mechanisms (such as neurogenic atrial fibrillation). Similarly, cerebrovascular relapses may imply a different pathological pathway than atrial fibrillation in this cohort. The real challenge, in my opinion, will be to be able to distinguish how many patients with AFDAS recognize a cardioembolic genesis of their neurological events. This aspect should be underlined and deepened in the discussion;

- another aspect to note is that only cases of atrial fibrillation diagnosed during the admission for the neurological event have been taken into account in the AFDAS group. As evidenced by many studies and as emerged with increasing strength thanks to the use of devices for the continuous monitoring of heart rhythm (such as implantable cardiac monitors or ECG-monitors for personal use), many patients with stroke have paroxysms of arrhythmia even at a distance from the index hospitalization. Therefore, many patients with AFDAS may have merged into the "non-AF" group. This should be highlighted within the limitations of the study and further discussed;

- it would also be interesting to investigate the incidence of major adverse cardiovascular events during the follow-up period in the three patient groups. Would it be possible to add this analysis to the study?

- The acronym AFADS does not sound right. I would suggest to modify it with AFDAS (Atrial Fibrillation Detected After Stroke), as widely reported in the literature. It also appears necessary, in conclusion, a minor revision of some typing and consecutio temporum errors of some sentences.

6. PLOS authors have the option to publish the peer review history of their article (what does this mean?). If published, this will include your full peer review and any attached files.

Reviewer #1: No

Reviewer #2: No

---

## [Author Response · Author response to Decision Letter 0]

27 May 2024

Response to Reviewers’ comments

Ref.: PONE-D-23-43796

Reviewers’ comments:

Must include:

Reply: Thank you. We had attached the files in including the rebuttal letter, marked-up and unmarked manuscripts, respectively.

Journal Requirements: When submitting your revision, we need you to address these additional requirements.

Reply: Thank you. Please see Manuscript_Clean file. We had revised the manuscript according to PLOS ONE's style.

2.We note that the grant information you provided in the ‘Funding Information’ and ‘Financial Disclosure’ sections do not match. When you resubmit, please ensure that you provide the correct grant numbers for the awards you received for your study in the ‘Funding Information’ section.

Reply: Thank you for your comment. We had added the ‘Funding Information’ section in the manuscript ((Please see Manuscript_Clean file P19L28-31)

3.Thank you for stating the following financial disclosure: This work was funded by grants from Tungs’ Taichung Metroharbor Hospital (protocol number TTMHH-109R0005). Please state what role the funders took in the study. If the funders had no role, please state: "The funders had no role in study design, data collection and analysis, decision to publish, or preparation of the manuscript." If this statement is not correct you must amend it as needed. Please include this amended Role of Funder statement in your cover letter; we will change the online submission form on your behalf.

Reply: Thank you. The funders had no role in this study. We had added the statement "The funders had no role in study design, data collection and analysis, decision to publish, or preparation of the manuscript." in the financial disclosure section (Please see Manuscript_Clean file P19L29-31).

4.Thank you for stating the following in the Acknowledgments Section of your manuscript: This work was funded by grants from Tungs’ Taichung Metroharbor Hospital (protocol number TTMHH-109R0005). We note that you have provided funding information that is not currently declared in your Funding Statement. However, funding information should not appear in the Acknowledgments section or other areas of your manuscript. We will only publish funding information present in the Funding Statement section of the online submission form. Please remove any funding-related text from the manuscript and let us know how you would like to update your Funding Statement. Currently, your Funding Statement reads as follows: This work was funded by grants from Tungs’ Taichung Metroharbor Hospital (protocol number TTMHH-109R0005).Please include your amended statements within your cover letter; we will change the online submission form on your behalf.

Reply: Thank you for your suggestions. Please see Manuscript_Clean file. We had moved the sentence “This work was funded by grants from Tungs’ Taichung Metroharbor Hospital (protocol number TTMHH-109R0005).” to Funding Statement.

5.When completing the data availability statement of the submission form, you indicated that you will make your data available on acceptance. We strongly recommend all authors decide on a data sharing plan before acceptance, as the process can be lengthy and hold up publication timelines. Please note that, though access restrictions are acceptable now, your entire data will need to be made freely accessible if your manuscript is accepted for publication. This policy applies to all data except where public deposition would breach compliance with the protocol approved by your research ethics board. If you are unable to adhere to our open data policy, please kindly revise your statement to explain your reasoning and we will seek the editor's input on an exemption. Please be assured that, once you have provided your new statement, the assessment of your exemption will not hold up the peer review process.

Reply: Thank you for your comment. We had appended the below sentences “Data are available from the National Health Insurance Research Database (NHIRD) published by Taiwan National Health Insurance (NHI) Bureau. Due to legal restrictions imposed by the government of Taiwan in relation to the “Personal Information Protection Act”, data cannot be made publicly available. Requests for data can be sent as a formal proposal to the NHIRD (https://dep.mohw.gov.tw/DOS/cp-5119-59201-113.html).” in the ‘Data Availability Statement’. (Please see Manuscript_Clean file P20L12-18)

6.We notice that your supplementary tables are included in the manuscript file. Please remove them and upload them with the file type 'Supporting Information'. Please ensure that each Supporting Information file has a legend listed in the manuscript after the references list.

Reply: Thank you. We had upload the S1 Table as a separated file with the file type 'Supporting Information'. 

 

Reviewer #1:

In this study, the authors retrospectively investigated more than 160.000 thousand patients from the National Health Insurance Research Database (NHIRD) of the NHI program in Taiwan. The aims of this study were to evaluate the clinical characteristics of newly diagnosed atrial fibrillation after a stroke (AFADS) in patients with ischemic stroke (IS) in a large population-based cohort; the differences in comorbidities and outcomes among the patients with known AF (KAF), AFADS, and without AF; the effects of anticoagulant treatment in patients with IS with either KAF or AFADS. The authors reported a very high mortality rate among patients with AF (both groups), and a higher risk of recurrent IS and hemorrhagic stroke for patients with KAF compared with AFADS.

The study is conducted in a large population and is overall well written, applying also a quite good statistical method. However, I have some major and minor comments:

Major

1) It is not clear at which time the data regarding OAC therapy has been extracted from the registry. Is it after the stroke events? Please provide this data.

Reply: Thank you for requesting this clarification. We apologize for not specifying the timing of the OAC prescription data extraction more clearly in our original submission. The data on OAC therapy for each patient was extracted from the National Health Insurance Research Database (NHIRD) based on prescriptions filled during the follow-up period after the index ischemic stroke hospitalization. Specifically, we captured all OAC prescriptions (warfarin or direct oral anticoagulants) for all patients from the time of their hospital discharge until the study end date, outcome event, or death. 

We have now clarified this in the Methods section ((Please see Manuscript Clean file P5L17-21), stating: "Antithrombotic treatments of study subjects were ascertained from NHI drug codes and Anatomical Therapeutic Chemical (ATC) codes. Antithrombotic prescriptions (including antiplatelet agents, warfarin and DOACs) were extracted during the follow-up period after the index stroke hospitalization until the study end date, outcome event, or death." We appreciate you pointing out this ambiguity, as specifying the timing of exposure ascertainment is critical for transparency and interpretability of our findings.

2) Following the previous point, many patients with KAF should have been already treated with OAC therapy before the stroke. I believe that it would be very interesting to know how many of them were already on treatment before the stroke.

Reply: Thank you for raising this important point. To address this, we re-analyzed the data to determine the proportion of patients with known AF (KAF) who received oral anticoagulant (OAC) treatment prior to their index stroke. We found that 7304 (47.2%) patients with KAF had received OACs treatment within the 6 months preceding the index stroke. Adherence to OAC therapy was evaluated using the proportion of days covered (PDC), calculated by dividing the days a patient had medication by the days in the observation period. Patients with a PDC of ≥80% were considered adherent. Over the 6 months and 30 days prior to the index stroke, 4462 (26.7%) and 4068 (24.4%) patients, respectively, were adherent to OAC therapy. 

We have added this new data to the Results (Please see Manuscript_Clean file P7L20-26) section: “In patients with KAF, 7304 (47.2%) patients had received OACs treatment within the 6 months preceding the index stroke. Adherence to OAC therapy was evaluated using the proportion of days covered (PDC), calculated by dividing the days a patient had medication by the days in the observation period. Patients with a PDC of ≥80% were considered adherent. Over the 6 months and 30 days prior to the index stroke, 4462 (26.7%) and 4068 (24.4%) patients, respectively, were adherent to OAC therapy.” 

We also highlight the concerning issue of OAC undertreatment in patients with KAF In the Discussion section: “ Only about a quarter of our patients with KAF were adherent to OAC therapy before the occurrence of the index stroke. Although PDC provides an indirect measure of treatment adherence, our findings suggest that a substantial proportion of KAF patients were not receiving guideline-recommended OAC therapy before their stroke events. The low rates of OAC prescription and adherence in this population underscore the urgent need for improved OAC initiation and adherence in clinical practice to prevent disabling strokes in high-risk patients with KAF (Please see Manuscript_Clean file P14L21-31). 

We appreciate you drawing attention to this crucial aspect of AF management. Thank you for this valuable suggestion.

3) Since data were collected regarding antiplatelet therapy, direct oral anticoagulants (DOAC) and vitamin K antagonist (VKAs), it would be interesting to see the results of the cox regression analysis also for the subgroups of antithrombotic treatment, especially in regard of hemorragic events.

Reply: Thank you for the comment. To investigate the impact of different antithrombotic therapies on the risks of outcome measures, we conducted multivariable Cox regression analyses comparing the hazard ratios (HRs) for hemorrhagic stroke associated with warfarin, direct oral anticoagulants (DOACs), antiplatelet therapy, and no antithrombotic treatment. The analyses were performed separately for all patients, those with known AF (KAF), those with AF diagnosed after stroke (AFADS), and those without AF (non-AF). In order to focus on hemorrhagic events, we summarized the results of hemorrhagic stroke in the Table listed below: 

Table. Summary of Multivariable Cox regression models to predict hemorrhage stroke among different antithrombotic treatments in different subgroups.

Hemorrhagic stroke Warfarin DOAC Antiplatelets No Antithrombitics

 All patients, n(%). 116(8.62) 487(4.80). 8118(6.65) 5435(17.75)

 HR (95%CI) *Ref. 0.57(0.46-0.70)** 0.81(0.67-0.97)* 3.71(3.08-4.46)**

 KAF, n (%) 41(9.95) 214(4.95) 286(9.18) 468(15.37)

 HR (95%CI) *Ref. 0.49(0.35-0.69)** 1.29(0.93-1.80) 4.75(3.41-6.61)**

 AFDAS, n(%) 14(10.22) 153(4.64) 190(11.54) 402(25.62)

 HR (95%CI) *Ref. 0.47(0.27-0.81)* 1.63(0.94-2.82) 8.33(4.84-14.34)**

 Non-AF, n(%) 61(7.66) 120(4.75) 7642(6.51) 4565(17.55)

 HR (95%CI) *Ref. 0.67(0.49-0.91)* 0.93(0.72-1.19). 4.16(3.23-5.36)**

KAF: known atrial fibrillation; AFDAS: atrial fibrillation detected after stroke; Non-AF: without atrial fibrillation; CI: confidence interval.

Ref.: reference.

#Adjusted age, sex, stroke severity index score, comorbidities (hypertension, diabetes, hyperlipidemia, coronary artery disease, heart failure, peripheral artery disease, chronic kidney disease, prior stroke/TIA), modified Charlson Comorbidity Index score.

*P<0.01; **P<0.001.

We present the main finding in the Results section: “DOACs were associated with significantly lower risks of hemorrhagic stroke compared to warfarin in KAF subgroup (HR 0.49, 95% CI: 0.35-0.69, p<0.001) and AFADS subgroup (HR 0.47, 95% CI: 0.27-0.81, p<0.001). The risks of hemorrhagic stroke were not significantly different between Warfarin and antiplatelet treatment in both KAF and AFDSA subgroups.” (Please see Manuscript_Clean file P8L23-27) to highlight the risks of hemorrhagic stroke related to different antithrombotic therapy for readers’ reference. We also add a brief statement in Discussion section: “ In addition, our results showed that DOACs have a superior safety profile with regard to hemorrhagic stroke risk compared to warfarin and antiplatelet therapy in patients with AF, whereas warfarin treatment was not associated with a higher risk of hemorrhagic stroke compared to antiplatelet treatment.” (Please see Manuscript_Clean file P12L15-19) 

Thank you for raising the important point which help us to have deeper understandings about the benefits and risks of different antithrombotic treatments in our patients. Currently, we could not answer some questions such as the reasons for OACs use in patients without AF, effects and risks of OACs in patients without AF, the reasons for with-holding antithrombotic treatment in different subgroups, and the risk of withholding antithrombotic treatment in different patient subgroups. The results inspire us to explore more important (and complex) issues in the future study.

4) Regarding the Cox regression analysis, the authors did not report about the adjustments made in the multivariable analysis for the risk of recurrent stroke, hemorragic stroke and all-cause mortality (Tables 2,3 and 4). Please provide this data.

Reply: Thank you for noting this omission in our reporting of the multivariable Cox regression analyses. We apologize for not clearly specifying the variables adjusted for in the models presented in Tables 2, 3, and 4.

To clarify, the multivariable Cox regression models for the outcomes of recurrent ischemic stroke (Table 2), hemorrhagic stroke (Table 3), and all-cause mortality (Table 4) adjusted for the following covariates: age, sex, stroke severity index (SSI) score, Comorbidities (including hypertension, diabetes mellitus, hyperlipidemia, coronary artery disease, heart failure, peripheral artery disease, chronic kidney disease, prior stroke/TIA), Modified Charlson Comorbidity Index (CCI) score, and anticoagulant treatment. These covariates were selected a priori for adjustment based on their potential to confound the associations between AF status and stroke outcomes or mortality. 

We have now clarified this in Methods section, under ‘Statistical Analysis’: “Multivariable Cox regression analyses were performed to estimate the HRs and 95% CIs for the association between AF subtype and the risk of recurrent ischemic stroke, hemorrhagic stroke, and all-cause mortality, adjusting for potential confounders including age, sex, stroke severity, comorbidities, and antithrombotic treatment.” (Please see Manuscript_Clean file P6L1-5), and added a footnote to each of Tables 2-4 stating: "Multivariable models adjusted for potential confounders including age, sex, stroke severity index score, comorbidities (hypertension, diabetes, hyperlipidemia, coronary artery disease, heart failure, peripheral artery disease, chronic kidney disease, prior stroke/TIA), modified Charlson Comorbidity Index score, and antithrombotic treatment."

Thank you for prompting us to clarify the rationale for covariate inclusio

---

## [Decision Letter · Decision Letter 1]

25 Jul 2024

Differences between atrial fibrillation diagnosed before and after stroke: a large real-world cohort study

PONE-D-23-43796R1

Dear Dr. Hsu,

We’re pleased to inform you that your manuscript has been judged scientifically suitable for publication and will be formally accepted for publication once it meets all outstanding technical requirements.

Kind regards,

Giulio Francesco Romiti

Academic Editor

PLOS ONE

Reviewers' comments:

Reviewer's Responses to Questions

6. Review Comments to the Author

Reviewer #1: The authors have addressed the points perviously reported in my peer review. I have no further comments.

Reviewer #2: Dear Authors, thank you for submitting your revised version of the manuscript.

The insights and changes made to the text and statistical analysis have increased the scientific value of the study.

The points that, in my opinion, required a major revision have been addressed.

I have no further comments or notes.

---

## [Editor Report · Acceptance letter]

6 Aug 2024

PONE-D-23-43796R1 

PLOS ONE

Dear Dr. Hsu, 

I'm pleased to inform you that your manuscript has been deemed suitable for publication in PLOS ONE. Congratulations! Your manuscript is now being handed over to our production team.

Kind regards, 

on behalf of

Dr. Giulio Francesco Romiti 

Academic Editor

PLOS ONE